# Eficacy of Cryotherapy in the Prevention of Oral Mucosistis in Adult Patients with Chemotherapy

**DOI:** 10.3390/ijerph18030994

**Published:** 2021-01-23

**Authors:** Ángel López-González, Marta García-Quintanilla, Carmen María Guerrero-Agenjo, Jaime López Tendero, Isabel María Guisado-Requena, Joseba Rabanales-Sotos

**Affiliations:** 1Department of Nursing, Physiotherapy and Occupational Therapy, Facultad de Enfermería, Universidad de Castilla-La Mancha, Campus Univesitario s/n, 02071 Albacete, Spain; angel.lopez@uclm.es (Á.L.-G.); gmartaq97@gmail.com (M.G.-Q.); joseba.rabanales@uclm.es (J.R.-S.); 2Group of Preventive Activities in the University Health Sciences Setting, Universidad de Castilla-La Mancha/UCLM, 13001 Ciudad Real, Spain; 3Castilla-La Mancha Health Service (Servicio de Salud de Castilla-La Mancha/SESCAM), Universidad de Castilla-La Mancha/UCLM, 02071 Albacete, Spain; carmenmga@hotmail.com (C.M.G.-A.); leman_1988@hotmail.com (J.L.T.)

**Keywords:** oral mucositis, patients’ cancer, oral mucositis prevention, oral cryotherapy

## Abstract

Oral mucositis (OM) is a common side effect of cancer therapies. It causes ulcerative, painful lesions in the oral cavity that can provoke malnutrition, increased risk of infection, longer hospital stays, and seriously affect the quality of life. Cooling the mucosa with oral cryotherapy (OC) during and/or after chemotherapy is the most accessible and tolerable intervention available. The aim of this study is to define the efficacy of OC for preventing OM induced by chemotherapy/radiotherapy in adult patients with cancer. Secondary endpoints include associated problems as pain. A systematic search was performed using the Pubmed, WOS (Web of Science), Cochrane Library, CINAHL, and BVS databases for articles published up to 2010. After inclusion and exclusion criteria were applied, a total of eight articles were analyzed in this review. In seven of the eight articles, the incidence of OM of all grades was significantly lower in the OC group compared with the no-OC group. Use of opioids and level of pain were also significantly reduced. OC is an effective intervention to reduce the incidence of OM induced by chemotherapy as well as the associated severity and pain. Based on these results, OC with only water or with chamomile, associated or not with other mouthwash therapies, is an effective intervention to reduce the incidence of OM induced by chemotherapy as well as the associated severity and pain.

## 1. Introduction

Oral mucositis (OM) is one of the most frequent complications suffered by patients affected by various types of cancer who are treated with chemotherapy, radiotherapy, or a combination of both. OM is defined as an alteration of the oral mucosa that causes inflammation and ulcerative lesions [1,2,3].

OM occurs in 20–40% of patients treated with conventional chemotherapy, in 80% of patients treated with high-dose chemotherapy prior to autologous bone marrow transplantation (HSCT), and even in almost all patients with head and neck cancer treated with radiotherapy combined with chemotherapy [1,2,3,4,5].

The physiopathogenesis of OM is a “cascade” process; the currently most accepted explanatory model describes it in five phases. In the first phase or “initiation phase”, cytotoxic agents or radiation cause DNA breakage of the epithelial basal cells, causing cell damage. This damage then leads to increased production of proinflammatory cytokines, which, in turn, induce cell death (transcription phase). At this point, the initial response to cell damage is magnified, and a positive feedback process is created in which more and more tissue is damaged (amplification phase). All this leads to the formation of ulcerative wounds and bacterial colonization, which overstimulates the inflammatory response, increasing the damage of the oral mucosa (ulceration phase). The process concludes in the healing phase after the treatment action is completed, the cells regenerate, and normal microbiota is restored [1,2,3,4,5,6].

The scale proposed by the WHO classifies it in five degrees: Grade 0, incipient and asymptomatic lesions; Grade I, oral soreness and erythema; Grade II, oral erythema, ulcers, solid diet tolerated; Grade III, oral ulcers, liquid diet only; Grade IV, oral alimentation impossible. On this scale, Grades III and IV correspond to severe OM [3].

The first manifestation of OM usually appears between the third and seventh day at the beginning of treatment; there is localized or generalized erythema, and this may be accompanied by stinging, the most affected tissues being the soft palate, the lateral edges of the tongue, the buccal mucosa, the tonsils, and the pharyngeal wall. In patients receiving chemotherapy for the treatment of solid tumors, OM may not progress to more severe stages due to the use of lower doses of chemotherapy/radiation. However, a large number of patients evolve and develop one or more ulcers, this being the period in which the patient experiences greater pain and discomfort. In the majority of those affected, it evolves into mucosal ulceration, this being the period in which the patient experiences the most pain and discomfort [7,8,9].

Severe OM is characterized by intense pain associated with ulcerative lesions as the main complication, originating a difficulty in feeding and a decrease in caloric contribution [2,6,7]. The inadequate nutrition will affect the immune system, and the ulcers are usually colonized by oral bacterial flora, very often by the herpes simplex and candida viruses. In patients who are immunosuppressed by chemotherapy, these lesions can be a route of entry for systemic infections, causing sepsis and potentially posing a danger to life [7]. It may become necessary to reduce and even suspend the doses of chemo/radiotherapy, with the worsening of the prognosis of the disease and the patient’s quality of life [6,7,10].

The main factors related to its appearance are related to the type and dose of cytostatic agents used. Regarding the type of agent, the most important are the antimetabolites (methotrexate, fluoronacil, cytarabine), which affect DNA synthesis and are associated with about 40–60% of OM incidences. In terms of dosage, it has been shown that the risk of suffering from OM increases with the intensity of the treatment, whatever the drugs chosen. Moreover, the planning of the cycles, their duration, the route of administration, localized radiotherapy in the head and neck, and the combination of chemo/radiotherapy are risk factors. Therefore, almost all patients who undergo an HSCT develop OM induced by the conditioning treatments [2,6,9].

Other possible factors such as age, type of tumor, poor oral health before and after treatment, malnutrition, alterations in the production and composition of saliva, and liver and kidney function could not be demonstrated in the appearance of OM [8,9].

The objectives of OM treatment are to prevent or reduce the severity of the lesions and to manage the associated symptoms, allowing the continuity of the cancer therapy [1,2].

The Multinational Association of Supportive Care and the International Society of Oral Oncology (MASC/ISOO) (Aurora, Ontario, Canada) have proposed, among the treatment methods for OM, basic oral care, application of growth factors and cytosines, anti-inflammatory agents, antimicrobials, protective agents, anesthetics and analgesics, laser and other phototherapies, natural agents and the application of oral cryotherapy (OC) [1,2,11].

OC consists of the local cooling of oral mucosal tissues using small pieces or sheets of ice, which, in the case of administration of chemotherapeutic agents, will produce vasoconstriction and decrease the distribution of the drug by the cells of the oral cavity. In the case of radiation therapy, the cold decreases the inflammation of the tissue reached by the radiation, which, in turn, reduces cell damage and prevents ulcerations [12,13,14].

OC is a normally well-tolerated intervention, with headache, tooth sensitivity, and numbness of the mouth being the most common side effects. In addition, it is the most accessible and efficient technique of all the proposals [15,16].

### Objectives: Peak Question

Due to the controversy about whether or not OC therapy is beneficial for patients with OM, this review aims to find out whether OC is effective in preventing severe OM and its influence on the onset or evolution of pain in adult cancer patients treated with chemotherapy.

## 2. Materials and Methods

### 2.1. Information Sources

The following review was conducted through a literature search that began in October 2019 and ended in January 2020 in the following health-related databases: Pubmed, WOS (Web of Science), Cochrane Library, CINHAL (Cumulative Index to Nursing and Allied Health Literature), and Biblioteca Virtual en Salud (BVS).

### 2.2. Search Strategy

The search strategy was based on the keyword search string in the MeSH/DeCS descriptors of the databases mentioned above. The string was filled in with the Boolean operators AND and OR.

Table 1 shows the PICO criteria used and how each part of the search string belongs to each PICO criteria, following the appropriate structure with its corresponding keywords.

### 2.3. Inclusion Criteria

The inclusion criteria included publication within the last 10 years (1 February 2010 through to 31 January 2020) and an adult patient population in which OC has been used to prevent or treat OM that appeared because of the treatment of their cancer with chemotherapy and/or radiation therapy.

The exclusion criteria were animal experimentation and OM produced by causes other than cancer treatment with chemotherapy or radiotherapy. Systematic reviews were excluded due to a lack of scientific quality.

### 2.4. Selection of Studies and Collection of Data

After an exhaustive search, a total of 368 results were obtained, of which 326 were eliminated by title and summary, leaving a total of 42 (Figure 1). Twenty-one duplicates were removed. The review was conducted independently by two researchers using the inclusion and exclusion criteria of this review.

Thus, a total of 21 articles were chosen for the literature review. However, 13 of them were discarded for not meeting the established inclusion criteria.

The results were structured under a standardized register using author, year, type of study, objective, randomization, blinded, country, duration of the study, patients who developed OM as a complication of cancer treated with chemotherapy and/or radiotherapy, and intervention.

### 2.5. Assessment of the Quality of Studies: Detection of Possible Bias

Rating scales were made to evaluate the quality of the studies. The PEDRO scale (11 items) was used for randomized controlled trials (RCTs) [17]. The selection criteria were taken into account if the study population was randomized and the study was blinded for the intervention. The groups were similar at the start of the study, and all subjects, therapists or those who collected the data, and evaluators were blinded as to whether there was a high proportion of population lost during the study.

In three observational case–control studies, the Newcastle–Ottawa scale for case–control was applied [18].

These scales were applied by one reviewer and analyzed by a second reviewer to detect possible biases.

All the articles included are characterized by a low risk of bias, according to the scales carried out.

### 2.6. Analysis of Data and Levels of Evidence

The degree of evidence depends on factors such as the type of study and the methodological quality. To evaluate the level of evidence, a qualitative assessment was carried out using the scale of the Agéncia dÁvaluació de Tecnología Médica [19]. According to this scale, all the selected RCTs are of high relevance since, as mentioned in the previous section, all of them have a low risk of bias and are observational case–control studies.

## 3. Results

After an exhaustive search of the different databases, a total of 368 results were obtained, of which 8 were finally selected to carry out this literature review. These articles were selected using the inclusion criteria. The populations studied come from Italy, Brazil, Turkey, Malaysia, Spain, Japan, and Canada.

The studies reviewed included between 38 and 140 participants, with a greater predominance of men in 7 of the eight articles reviewed [16,17,19,20,21,22,23]. All of them included adult patients, ranging in age from 18 to over 70 years. Table 2 and Table 3 show the characteristics and results of the selected studies.

In four of the studies, patients underwent HSCT with high-dose chemotherapy [20,23,24,25] while in the other four studies, the patients suffered from cancer of the digestive system (colorectal, stomach, esophagus, and pancreas) [21,22,24,26]. The study periods were from four months to seven years in duration. All selected studies focused on OM produced by chemotherapy treatments. Chemotherapy regimens were mainly based on melphalan in cases of HSCT [20,23,25,27] and fluoronacil in cases of digestive cancer [21,24,26]. Regarding the design of the studies, in all of them, the experimental group (EG), formed by the patients who were treated with OC, was compared with a control group (CG). In four of the articles, patients who received standard oral hygiene were assigned the CG [20,25,26,27]. In another three articles, OC was compared to the effects of other types of treatments for OM applied to the CG, such as chlorhexidine rinses [22], bicarbonate rinses [24], and the use of Caphosol^®^ (EUSA Pharma (Europe) Limited – Stevenage, UK) [23]. In addition, a variant of OC made with chamomile infusion was analyzed in the remaining study [21].

## 4. Discussion

OM is one of the adverse effects of chemotherapy and radiotherapy that mostly worsens the quality of life of cancer patients in addition to increasing hospitalizations and financial expenses. There are many treatments that have tried to reduce both its incidence and its severity, including OC.

The present literature review confirms the benefits of OC in the control of OM produced as an adverse effect of chemotherapy treatment in cancer patients, as well as being cheaper, more accessible, and better tolerated by patients.

### 4.1. OC in OM Prevention

The effectiveness of OC was demonstrated in six of the articles, where the incidence and severity of OM were found to be significantly lower in the EG than in the CG, to which it was not applied [20,23,24,25,26,27]. Only one of them proposed the use of chlorhexidine rinses instead of OC to decrease the incidence of OM and to facilitate an oral diet [22]. In the remaining study, it was proposed to replace conventional ice with ice cubes made with chamomile infusion since the incidence of OM and the pain perceived by the patients in that EG were lower [21].

Additionally, the occurrence of severe OM was studied in five articles, and in all of them, a significant reduction was observed in the groups to which OM was applied, even to the point of never occurring [20,24,25,26,27].

### 4.2. Influence of OC on the Occurrence of Pain in Adult Cancer Patients Treated with Chemotherapy

Pain is a variable that was taken into account in six of the studies, measured based on the need for the use of opioids or through rating scales [20,21,23,24,25,27]. In the first case, Marchesi et al. and Chen et al. reported that the administration of opioid analgesics was significantly lower in the EG [20,27]; however, Batlle et al. found no difference between the EG and the CG [25]. In terms of direct pain assessment, in two of the articles, patients treated with OC expressed lower scores than those who were not [21,24]. Furthermore, in the study by Svanberg et al., it was observed that the association of OC + Caphosol^®^ offered no additional effect on pain compared to using OC alone [23].

Finally, seven of the eight selected articles support the use of OC, including a camomile-infused variant, as an effective measure to decrease its incidence and severity [20,21,23,24,25,26,27]. Finally, in all of them, the treatment was well-tolerated in its great majority and was without adverse effects.

### 4.3. Strengths and Limitations

The present review presents some limitations, such as a sample that does not allow the generalization of the results found and the variability of the types of OC used.

Among the strengths is the recent research included (from the last 10 years), so the information is current. In addition, the results of the studies are similar, and the quality of all of them is high. Moreover, the great variability of the countries where the studies have been carried out provides a global vision.

Due to the scarcity of articles on the effect of OC in the prevention of OM and its main complications, such as pain, more research should be carried out with larger population samples to obtain conclusive data.

### 4.4. Implications for Clinical Practice

OC is an effective treatment to decrease the incidence of OM in patients receiving chemotherapy/radiotherapy. It is also effective in preventing the progression of more severe phases of ulceration.

OC decreases pain in patients with OM that has developed as a result of their cancer treatment with chemotherapy/radiotherapy.

It is a very safe therapeutic option due to the great tolerance of the patients and the scarcity of adverse effects, besides being a very cheap and affordable resource for any institution.

## 5. Conclusions

After analyzing the selected articles, we conclude that OC is an effective treatment to prevent the appearance of oral OM in patients who are being treated for cancer with chemotherapy/radiotherapy. OC application avoids the worsening of OM to a more serious phase of ulceration and alleviates its main symptom, pain.

OC is a therapeutic option that has been shown to be safe for patients, with a high tolerance level, given the scarcity of adverse effects. OC is a very economical resource that is affordable for any institution, showing itself as a highly efficient therapeutic option.

OM is a very widespread complication with a high impact on the quality of life of patients.

## Figures and Tables

**Figure 1 ijerph-18-00994-f001:**
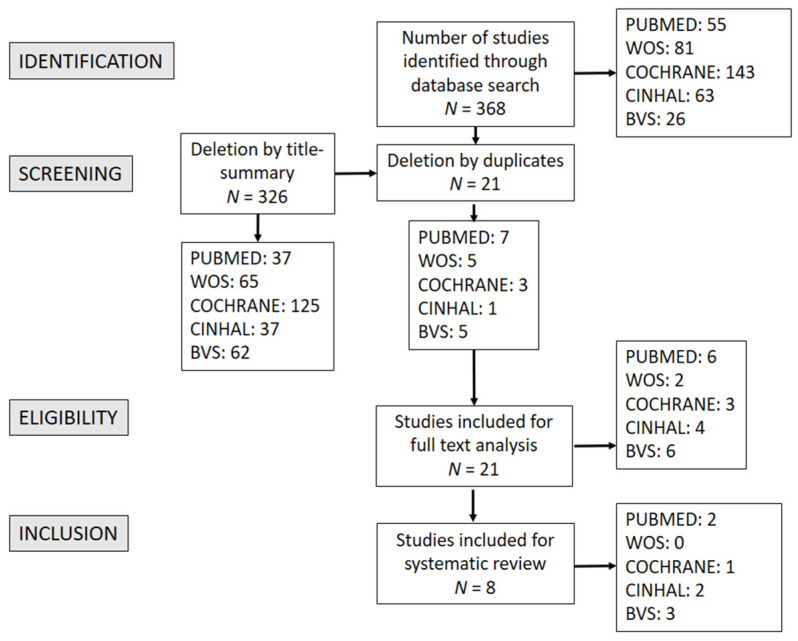
Prisma flow chart.

**Table 1 ijerph-18-00994-t001:** PICO criterion.

Criterion (PICO)	Keywords
Population (P)	(“patients’ cancer”) and (“oral mucositis”) and (“chemotherapy”) and (“radiotherapy”)
Intervention (I)	oral cryotherapy
Outcome (O)	OM prevention OR patient´s benefits

**Table 2 ijerph-18-00994-t002:** Description of the type of study, intervention, sample size, and year of the selected studies.

Author—Year	Type of Study	Type of Intervention	Sample Size
Marchesi F. et al., 2017 [20]	Randomization method: it is based on two groups (experimental and control). Place: Hospital Sant’Eugenio in Roma (Italy). Participants were not blinded. All patients signed a written informed consent, and the study was approved by the Ethical Committee, in compliance with the Helsinki declaration.	From October 2013 to January 2016, patients were enrolled in this study. CG: oral standard care. EG: after IV melphalan administration, on day-2, patients in EG received ice chips with rounded corners in their mouth during chemotherapy infusion. When the ice melted, it was immediately replaced. OM was assessed daily using the National Cancer Institute Common Toxicity Criteria version 4.02. Patient pain due to OM was monitored daily using the numerical rating scale. In case of severe and uncontrolled pain, IV opioids were administered. All patients (EG and CG) underwent uniform anti-infectious and support therapy (ciprofloxacin, cotrimoxazole, and valaciclovir). All patients (EG and CG) underwent mouth rinses with oral nystatin-based protocols three times daily. Patients were hospitalized and remained as inpatients for the duration of the study. No data on how long patients received chemotherapy or how long after therapy when some degree of OM appeared.	*N* = 72 (36 EG, 36 CG), 46 males, 26 females. Inclusion criteria: Patients aged ≥18 years, with multiple myeloma undergoing HSCT after being treated with high doses of melphalan. Exclusion criteria: Patients who had experienced previous episodes of OM or with previous exposure to chemotherapy or neck/head radiotherapy were excluded. Characteristics: age (EG: 58 ± 13.5; CG: 56 ± 17).
Diniz et al., 2016 [21]	Randomization method: it is based on two groups (experimental and control). Place: Hospital Center of High Complexity Oncology del Hospital Universitario en Brasilia (Brasil). Only the doctor was blinded to randomization. Patients were not possible to blind. All patients provided written consent prior to starting the procedures. The study was approved by the Committee on Ethics Research of the School of Health Sciences of the University of Brasília.	Between March 2012 and March 2015, patients were invited to participate in the study. The study interventions were performed only during the first 5 days of the first cycle of chemotherapy. Patients who agreed to participate in the study watched a video explaining how to perform oral hygiene and received an oral hygiene kit (toothbrush, nonabrasive toothpaste, and dental floss). During chemotherapy treatment, the following were administered prophylactically: CG: OC with only water was administered prophylactically. EG: Patients in the control group received a cup of ice chips made with pure water, while patients in the chamomile group received a cup of ice chips made with chamomile infusion at 2.5%. Both groups were instructed to swish the ice around in their oral cavity for at least 30 min, starting 5 min before the chemotherapy infusion. During the intervention, patients were asked to fill a questionnaire about ice taste, discomfort, and pain regarding cryotherapy. A doctor evaluated the oral mucosa on days 8, 15, and 22 after the first chemotherapy infusion. Patients underwent four to six courses of chemotherapy, each consisting of five consecutive days of chemotherapy infusion, followed by 21 days of rest. The study interventions were performed only during the 5 days of the first course. No data showing how long after therapy before some degree of OM appeared.	*N* = 38 (20 EG—11 males, 9 females, 18 CG—9 males, 9 females). Inclusion criteria: elderly patients with gastric or colorectal cancer who received ambulatory IV chemotherapy for the first time (5-fluorouracil and leucovorin). Intact and healthy oral mucosa, without dental problems, and without a history of hypersensibility or adverse reaction to chamomile or any plant of the Asteraceae or Compositae family. Characteristics: age EG: 54.7 (SD 8.15); CG: 55.2 (SD 9.5).
Erden et al., 2017 [22]	Randomization method: it is based on three groups (2 experimental and 1 control). Place: Erzurum Ataturk University Research and Application Hospital in Erzurum (Turkey). Participants were not blinded. Oral consent was obtained from the patients for their participation in the study before the questionnaire forms were administered. The patients were also informed verbally about the study. Participation was voluntary, and the patients could withdraw from the study at any time without giving a reason. Approval was obtained from the Ethics Committee of the Medical Faculty, Ataturk University.	The observation period was 15 days for each participant. During chemotherapy treatment, the following were administered prophylactically: EG1: received chlorhexidine mouthwash. EG2: OC with only water was administered prophylactically. CG: standard oral care. EG1 data were collected from patients in the study groups who had oral care with chlorhexidine twice a day; EG2 had cryotherapy once a day. Chlorhexidine mouthwash was applied six times a day to patients with Grade III oral mucositis, and it was applied eight times a day to patients with Grade IV oral mucositis. Tooth brushing is not recommended to patients with Grade III and IV oral mucositis because of possible ulcerations due to physical irritation. CG data were collected from patients who followed standard oral care protocol (washing with plenty of water mouthwash). The duration of the patient’s disease was 4–9 or more months, and the duration of cancer therapy was 1–9 months. No data showing how long patients received chemotherapy or how longafter chemotherapy before some degree of OM appeared.	*N* = 90 (30 EG1, 30 EG2, 30 CG). Inclusion criteria: All subjects had Grade III–IV oral mucositis due to chemotherapy received for various types of cancer, and all of them were unable to take food orally. Exclusion criteria were not named.
Svanberg et al., 2015 [23]	Randomization method: it is based on three groups (2 experimental and 1 control). Place: Akademiska University Hospital in Uppsala (Sweden). Participants were not blinded. The study was approved by the regional Research Ethics Committee.	From September 2010 and October 2011, patients were enrolled in this study. During chemotherapy treatment, the following were administered prophylactically: CG: OC with only water was administered prophylactically. EG: OC + Caphosol^®^ was administered prophylactically. OC was given in the form of ice cubes or crushed ice to be kept in the mouth during the actual infusion of the HSCT. Thirty (30) mL Caphosol^®^ was administrated for rinsing the whole oral cavity four times/day, starting prior to HSCT and ending on day 21. All patients received intravenous conditioning chemotherapy based on diagnosis. The average duration of chemotherapy treatment, depending on the type of cancer, was 4 days. No data showing how long before some degree of OM appeared.	*N* = 40 (20 EG—11 males, 9 females, 18 CG—9 males, 9 females). Inclusion criteria: patients >16 years, with various types of cancer (acute myeloid leukemia, acute lymphoblastic leukemia, chronic lymphatic leukemia, chronic myeloid leukemia, chronic myelomonocytic leukemia, myelodysplastic syndrome), undergoing HSCT after being treated with chemotherapy and radiotherapy. All patients received intravenous conditioning chemotherapy and (when required) total body irradiation on the basis of diagnosis. Exclusion criteria were not named. Characteristics: age between 17 and 67 years (*p* > 0.05; EG: 50.4 ± 10.6; CG: 59.6 ± 13.2)
Idayu et al., 2018 [24]	Randomization method in two groups: experimentation and control. Place: University of Malaya Medical Centre in Kuala Lumpur (Malaysia). Participants were not blinded. All patients provided informed consent and were ensured of the confidentiality of their participation. The study was approved by the Ethical Committee.	During treatment with chemotherapy (fluorounacil), the following was administered prophylactically: CG: standard care and rinses with sodium bicarbonate were administered prophylactically. EG: were given ice chips to hold in their mouths for 30 min during chemotherapy administration (the ice chips were replenished as they melted, and patients were instructed to move the ice in an attempt to keep the entire oral cavity cold), followed by sodium bicarbonate mouthwash (three times daily) postchemotherapy until the next cycle. No data showing the duration of the study, how long patients received chemotherapy, or how long after therapy before some degree of OM appeared	*N* = 80 (40 EG—22 men, 18 women, 40 CG—23 men, 17 women) Inclusion criteria: patients >20 years old with colorectal cancer, scheduled for fluorouracil-based chemotherapy at their first cycle of chemotherapy. According to the Eilers scale (1988), only patients with scores ranging from 1–8 were recruited into the study. Characteristics: age: between 20 and 60 years (mean = 48.4, SD = 9.2; *p* > 0.05).
Batlle et al., 2014 [25]	Retrospective cohort study. Randomization method in two groups: experimentation and control. Location: not specified.	From August 2006 to July 2011, 134 consecutive patients were enrolled in the study. All participants underwent an oral care protocol consisting of a sodium bicarbonate mouthwash from day 7 of HSCT to hospital discharge. CG: standard care and rinses with sodium bicarbonate were administered prophylactically. EG: OC consisted of ice chips sucked on before infusion (10 min), during infusion (15 min), and after chemotherapy administration for a total of 40 min. and then mouthwashes with sodium bicarbonate up to hospital discharge. The mean time to the initiation and the duration of OM did not differ among groups (2.50 and 9.20 days, respectively). No data on how long the patients received chemotherapy.	*N* = 134 (66 EG—47 men, 19 women, 68 CG—39 men, 29 women) (*p* = 0.094). Inclusion criteria: patients >20 years old, with hematological cancer treated with chemotherapy (high-dose melphalan (HDmel)) and HSCT. Characteristics: age between 23 and 70 years old. EG: 56 (23–69), CG: 55 (25–70) (*p* > 0.05).
Okamoto et al., 2019 [26]	Retrospective cohort study. Randomization method in two groups: experimentation and control. Place: Department of Gastroenterological Surgery, Kanazawa University (Japan). All procedures followed were in accordance with the ethical standards of the responsible committee on human experimentation and the Helsinki Declaration of 1964 and later versions. Informed consent or a substitute for it was obtained from all patients included in the study.	From March 2011 and July 2016, patients were enrolled in this study. CG: did not receive OC. EG: OC performed routinely for patients receiving chemotherapy. The patients were instructed to suck continuously on several pieces of ice from 10 min before until after the end of the chemotherapy infusion. The chemotherapy (cisplatin, 5-fluorouracil) was administered for 1–5 days, repeated every 4 weeks. No data showing how long before some degree of OM appeared.	*N* = 72 (58 EG—50 men, 8 women; CG 14—12 men, 2 women) (*p* > 0.999) Inclusion criteria: patients with primary esophageal cancer staged according to the 7th edition of the tumor-node-metastasis (TNM) classification of malignant tumors and treated with chemotherapy prior to surgery. Characteristics: age between 57 and 71 years old. EG: 64.0 (57.0–69.0). CG: 64.5 (57.3–70.5) (*p* > 0.05).
Chen et al., 2017 [27]	Retrospective cohort study. Randomization method in two groups: experimentation and control. Place: Victoria Hospital, London Health Sciences Center (UK). Participants were not blinded. The study protocol was approved by the Office of Research Ethics at the University of Western Ontario, Lawson Health Research Institute. Due to the retrospective nature of the study, no patient consent was required.	The study examined patients over the span of seven years, from 2006 to 2013. Medical charts of consecutive patients with multiple myeloma undergoing autologous HSCT, admitted over the period of 2006 to 2013, were reviewed. Two groups of patients were compared in this study analysis: CG: patients treated with chemotherapy between 2007 and 2009 who did not receive OC until 2010. EG: OC performed routinely for patients receiving chemotherapy from 2010 to 2013. Patients were instructed to hold ice chips in their mouth for 5 min prior to high-dose melphalan infusion, during the 30-min infusion, and 30 min after completion of the infusion. The conditioning regimen of chemotherapy (high-dose melphalan) was administered 2 days before the transplant date. No data showing how long before some degree of OM appeared. The duration of OM in mean days was 10.1–7.8 (SD 4.9 ± 6.2, respectively).	*N* = 140 (70 EG—56 men, 14 women; CG 70—38 men, 22 women) (*p* = 0.01). Inclusion criteria: patients >18 years old, with lymphoma and multiple myeloma, receiving high-dose chemotherapy (mephalan) for HSCT. Characteristics: age between 57 and 71 years old. EG: 53.5 (±7.5),; CG: 56.5 (±7.3) (*p* = 0.02).

**Table 3 ijerph-18-00994-t003:** Description of the results, conclusions, quality, and year of the selected studies.

Author—Year	Results	Conclusions	Quality
Marchesi F et al., 2017 [20]	Occurrence of Grade III–IV OM (%): EG: 5.6; CG: 44 (*p* = 0.0002). Occurrence of any grade OM (%): EG: 16; CG: 58.3 (*p* = 0.001). Need for opioid IV therapy (%): EG: 2.8; CG: 33.3 (*p* = 0.001).	The study provided relevant data to support OC during HDM administration as the standard of care in preventing OM in myeloma patients undergoing HSCT.	9/11
Diniz et al., 2016 [21]	EG patients never presented OM ≥ Grade 2 (*p* > 0.005). OM appearance in any degree: EG: 30%; CG: 50% (*p* = 0.01). Mouth pain: EG had less mouth pain than CG (*p* > 0.05).	The infusion of OC with chamomile reduces the appearance of OM compared to OC with water alone; it also reduces mucosal pain. The occurrence of OM was lower in patients who used OC made with chamomile infusion than in patients who used OC made only with water. When compared to the control group, the chamomile group presented less mouth pain and had no ulcerations. OC was well tolerated by both groups, and no toxicity was identified.	9/11
Erden et al., 2017 [22]	Oral nutrition transition time in days (mean ± SD) EG1: 8.53 ± 1.04_._ There was a statistical difference between experimental group 1 and the control group (*p* < 0.01). EG2: 12.13 ± 1.81_._ There was no statistical difference between experimental group 2 and the control group (*p* > 0.05). CG: 13.53 ± 1.69_._ There was no statistical difference between experimental group 2 and the control group (*p* > 0.05).	The analysis of this study showed that the transition time of oral nutrition of the patients in the experimental group that applied chlorhexidine was shorter than the transition time of oral nutrition of the patients of the group that applied OC; in both experimental groups (EG1 and EG2), transitional time was shorter than the control group. Parallel to this finding, it was found that the degree of OM was reduced. According to this result, using chlorhexidine or OC mouthwash for the prevention and treatment of oral mucosis should be offered.	8/11
Svanberg et al., 2015 [23]	There is no difference between EG and CG in the degree of OM at day 21 of treatment (2.45 vs. 2.30 on average, according to the WHO scale; *p* > 0.05). The perception of oral pain in both groups, assessed with the visual analog scale, showed no difference (3.55 vs. 2.7 on the visual analog scale; *p* > 0.05).	No additional significant effect of combining Caphosol^®^ with OC in the prevention and treatment of OM.	9/11
Idayu et al., 2018 [24]	The appearance of OM was as follows (*p* < 0.05): EG: 29 participants did not present OM (Grade 0), 11 did (Grade I); CG: 38 participants presented OM of Grade II or higher. In the EG, 27 indicated no pain, while 38 of the CG indicated moderate to severe pain. Pain associated with OM (*p* > 0.005): EG: 27 participants reported no pain; CG: 18 participants reported moderate pain and 20 participants reported severe pain.	OC followed by bicarbonate-based mouthwash could help prevent oral mucositis and pain. This finding helps shed light on evidence supporting the use of oral cryotherapy, which is cost-effective and has few side effects, as a preventive strategy. OC is easily implemented in tandem with the use of a sodium bicarbonate mouthwash. The potential benefit of cryotherapy in the prevention of oral mucositis and the associated pain appears to improve the quality of life of patients undergoing fluorouracil-based chemotherapy.	8/11
Batlle et al., 2014 [25]	Population who developed OM to any degree: EG: 44% vs. CG 82% (*p* < 0.001). The incidence of OM Grades III and IV: EG: 15% vs. CG: 31% (*p* = 0.031). Opiates were required in EG: 10% and CG: 15% (*p* = 0.305).	The authors indicated although this was a nonrandomized study and the conditioning regimens were not homogeneous, OC reduced the severity of OM in patients treated with regimens compared with saline rinses. Additionally, OC was cost-effective and well tolerated by the patients. In summary, OC represents an effective and inexpensive supportive measure to prevent OM induced by HDmel-based regimens.	7/11
Okamoto et al., 2019 [26]	The incidences of OM Grades I–III was 71.4% (CG) and 24.1% (EG); *p* = 0.001. The incidence of Grade III OM was also lower: EG 0% vs. CG 28.6%; *p* = 0.001.	OC may be a useful prophylactic approach for chemotherapy-induced OM in patients with esophageal cancer.	8/11
Chen et al., 2017 [27]	The incidence of OM was significantly lower in the EG than in the CG (71.4% vs. 95.7%; *p* < 0.001). The mean degree of OM in the CG vs. the EG was higher (2.5 vs. 2; *p* = 0.03. The use of parenteral analgesics was significantly lower in the EG (25.7%) than in the CG (44.2%), *p* = 0.02.	OC protocol implemented at HSCT resulted in significantly lower incidences and severity of oral mucositis. The mean duration of oral mucositis experienced by patients was shortened, and the need for the use of parenteral narcotics was decreased as well. These results provide evidence for the continued use of oral cryotherapy, an inexpensive and generally well-tolerated practice, in patients receiving high-dose melphalan for autologous HSCT.	8/11

## Data Availability

The study did not report any data.

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
