# Peer review of "Eficacy of Cryotherapy in the Prevention of Oral Mucosistis in Adult Patients with Chemotherapy"

_ijerph, 2021, doi:10.3390/ijerph18030994_

Round 1
Reviewer 1 Report
Thank you very much for inviting me to review this systematic review.
Cryotherapy is an inexpensive intervention with no major adverse effects that is commonly used to prevent oral mucositis.The use of cryotherapy for the prevention and treatment of mucositis is a topic frequently reviewed by international institutions such as Mucositis Study Group of the Multinational Association of Supportive Care in Cancer/International Society of Oral Oncology (MASCC/ISOO). These associations carried out periodically excellent systematic reviews about these interventions, so it is difficult for other studies to conclude anything different.
After reviewing your study I think it is more than appropriate to review the methodology followed for it. I will now comment on a series of considerations that must be evaluated:
-I do not think it is correct to include only studies from 2010 onwards. There are good and important studies done on this subject in the past.
-In the introduction you focus mainly on the physiopathological mechanism of oral mucositis after oncological treatment. I think you should summarize that part and focus primarily on prevention and treatment which is what you are going to review.
-It would be important to review the use of cryotherapy in the patient receiving chemotherapy treatment for oral cancer. According to different studies these patients should not used that treatment because it would diminish the effectiveness of chemotherapy treatment for this type of cancer.
-Review the wording of the objective.
-The search in the databases is not completely correct. You are missing important terms such as "chemotherapy", "head and neck radiotherapy", "oral mucositis prevention", "oral mucositis treatment", "oral pain", ...
-Review your PICO question (Patient, Intervention, comparison, outcome). With regard to patients, it does not specify what type of cancer patient you will include. This point should be developed: patient who is going to receive chemotherapy and/or head and neck radiotherapy, within the chemotherapy and radiotherapy it would be advisable to specify the treatment and the dose. With respect to the intervention, the time of application, before, during or after the treatment, should be specified. You do not comment the comparison and I can see later that you include studies that compare cryotherapy with other treatments. And you must also specify how you will measure the result, which mucositis scales you will include, how you will measure the pain,...
-I also wanted to comment on the fact that it is not true that systematic reviews have no quality as a type of study. In fact, you are conducting one.
-Nor does it specify what type of studies it will include: RCTs, case-control, observational studies... This must also be specified.
-The methodological quality of the RCTs is usually evaluated with the Cochrane Collaboration tool.
I think that before evaluating the results and the discussion, all these points should be reviewed in order to assess the quality of the results. Similarly, I believe that the results should be given according to the patients included (treatments received from chemotherapy, head and neck radiotherapy, or both) in order to correctly systematize the results.
Author Response
We would like to thank you for giving us the opportunity to review our manuscript, and also want to thank the reviewer´s comments and suggestions

Reviewer 2 Report
Given the incidence of neoplastic diseases, the Authors focused on an important topic related to the reduction of the side effects of radio or chemotherapy. However, the work is presented in a rather chaotic manner, and the description of including and excluding the analyzed publications takes more than comparing them.
Unfortunately, the presented table, instead of simplifying the information, complicates it very much, because it describes the methods used very generally and randomly. There is no detailed information about the research carried out. Only in one case [26] is the description of the OC carried out more detailed and therefore legible. Perhaps two tables should be created and the information contained therein should be divided, e.g. the demographic characteristics of the respondents should be described in several columns.
After reading the manuscript, you get the impression that the table and the text are completely separate works and have little in common.
The article lacks a summary, and the one that is presented is very laconic, e.g. line 162 - what is the description of the duration of the study about? How long did patients receive chemotherapy? After what time did OM appear? Were patients presented with the protocol for the management of the study? None of this information is included in the article.
Lines 215-217 - the information contained in the table should also be included in the text, i.e. which research allowed for effective in preventing of OM? Due to the very small database (8 articles), the comparison should be presented in more detail.
Author Response

(The authors gave the same response as above.)

Reviewer 3 Report
Thank you for sending me the manuscript titled ''Efficacy of Cryotherapy in the Prevention of Oral Mucositis in Adult Patients with Chemotherapy". The study shows quality and originality.
The manuscript would benefit from professional English language revision. Words such as gender and thorough should replace sexes and exhaustive respectively. Also, ''tried to try to reduce'' should be revised on line 179 p:12. Single numbers should not be used in digits in a research piece of work where it should be written in words. The outcome in table 1 PICO criteria should be revised as ''patients' prevention'' does not make sense, prevention from what?? do you mean OM prevention?
Authors need to make it clear whether it is systematic review or narrative review or literature review. This is because in the Discussion and Results sections the phraise ''systematic review'' was used to describe the work, whilst systematic reviews have different protocol settings from the current study's objectives. Perhaps adding ''Narrative Review'' to the title will make it clearer.
Author Response

(The authors gave the same response as above.)

Round 2
Reviewer 2 Report
The Authors made appropriate corrections. I have no more comments.